# A Review of SAW-Based Micro- and Nanoparticle Manipulation in Microfluidics

**DOI:** 10.3390/s25051577

**Published:** 2025-03-04

**Authors:** Débora Amorim, Patrícia C. Sousa, Carlos Abreu, Susana O. Catarino

**Affiliations:** 1Microelectromechanical Systems Research Unit (CMEMS), School of Engineering, Campus de Azurém, University of Minho, 4800-058 Guimarães, Portugal; deboram@ipvc.pt (D.A.); cabreu@estg.ipvc.pt (C.A.); 2INL—International Iberian Nanotechnology Laboratory, 4715-330 Braga, Portugal; patricia.sousa@inl.int; 3LABBELS—Associate Laboratory, 4800-122 Braga, Portugal; 4ADiT-Lab—Instituto Politécnico de Viana do Castelo, 4900-367 Viana do Castelo, Portugal

**Keywords:** acoustofluidics, interdigitated transducers, microfluidics, piezoelectricity, separation, surface acoustic waves

## Abstract

Surface acoustic wave (SAW)-based microfluidics has emerged as a promising technology for precisely manipulating particles and cells at the micro- and nanoscales. Acoustofluidic devices offer advantages such as low energy consumption, high throughput, and label-free operation, making them suitable for particle manipulation tasks including pumping, mixing, sorting, and separation. In this review, we provide an overview and discussion of recent advancements in SAW-based microfluidic devices for micro- and nanoparticle manipulation. Through a thorough investigation of the literature, we explore interdigitated transducer designs, materials, fabrication techniques, microfluidic channel properties, and SAW operational modes of acoustofluidic devices. SAW-based actuators are mainly based on lithium niobate piezoelectric transducers, with a plethora of wavelengths, microfluidic dimensions, and transducer configurations, applied for different fluid manipulation methods: mixing, sorting, and separation. We observed the accuracy of particle sorting across different size ranges and discussed different alternative device configurations to enhance sensitivity. Additionally, the collected data show the successful implementation of SAW devices in real-world applications in medical diagnostics and environmental monitoring. By critically analyzing different approaches, we identified common trends, challenges, and potential areas for improvement in SAW-based microfluidics. Furthermore, we discuss the current state-of-the-art and opportunities for further research and development in this field.

## 1. Introduction

Particle and cell manipulation in microfluidics represents a dynamic and rapidly evolving field with profound implications across diverse sectors, including healthcare, biotechnology, and environmental sciences [1]. Microfluidics involves the precise manipulation of fluids at the microscale, offering unique advantages such as high automation, minimal human interaction, low cost, and easy integration and miniaturization [2]. Techniques employed in microfluidics can be broadly categorized as passive and active. Passive techniques harness inherent fluidic properties and hydrodynamic forces for particle and cell manipulation, offering simpler equipment setups. In contrast, active techniques rely on external fields such as magnetic, optical, acoustic, electric, and thermal to exert control over particles and cells, offering greater flexibility, accuracy, and throughput [3,4,5,6]. 

Among these techniques, acoustics stands out for its low consumption, high throughput, label-free, non-contact, and non-invasive approaches. These benefits have contributed to the exponential growth in the development of systems integrating acoustic techniques in microfluidics, a field widely known as acoustofluidics. Acoustofluidic devices can manipulate objects at micro- or nanoscales based on size, density, or mechanical properties, such as compressibility or deformability [7]. These systems offer numerous benefits, including increased precision in separation, low power, minimal sample volume requirements, reduced costs, as well as the potential for disposability [1]. Due to these characteristics, acoustofluidics has been widely employed in recent research for the manipulation of flows (fluid pumping, mixing, and separation [8,9]), and micro- and nanoparticles. Examples of bioparticles’ manipulation found in the literature include microfluidic-based isolation of circulating tumor cells [10,11,12,13], separation of *Escherichia coli* bacteria [13,14,15], separation of red and white blood cells, manipulation of DNA segments and exosomes [12,16], and separation of *Thalassiosira eccentrica* diatoms [17].

Beyond their applications as actuators, SAW devices also hold great potential as sensors in microfluidics. The interaction of acoustic waves with the medium can be exploited for the highly sensitive detection of physicochemical properties such as mass [18], viscosity, density [19], and biomolecular concentrations {Citation}. SAW sensors typically operate from around 100 MHz to beyond 10 GHz, and output results by measuring variations in the wave velocity or attenuation caused by changes on the surface of the piezoelectric substrate, making them ideal for biosensing, chemical reaction monitoring, and pathogen detection [20,21].

In acoustofluidics phenomena, fluids and particles are manipulated by sound waves, which operate based on two major mechanisms, depending on their generation and propagation methods: bulk acoustic waves (BAWs) and surface acoustic waves (SAWs) [22,23,24]. Both require a piezoelectric substrate for propagation of the acoustic wave, but while BAWs propagate through the entire thickness of the material, SAWs are limited to the surface and travel along the substrate with minimal energy loss. Consequently, SAW microfluidic platforms typically require less energy to achieve the same acoustic effect as BAW-based platforms. Furthermore, BAWs are suitable for manipulating larger particles, whereas SAWs allow for precise manipulation at the micro- and nanoscales, offering high spatial resolution [25]. By modulating the frequency and amplitude of the SAWs, localized acoustic forces can be exerted on particles or cells, enabling precise manipulation and sorting within microfluidic channels [2,3,25,26]. 

Particles’ size plays a crucial role in determining manipulation characteristics. Larger particles, typically in the range of 10–100 µm, are more easily influenced by acoustic forces, making them suitable for efficient trapping, separation, and concentration within microfluidic channels. For instance, most of the biological cells, which range from 5 to 50 µm, can be effectively manipulated using acoustofluidic devices. On the other hand, in medical applications, particles such as blood cells (6–8 µm) and exosomes (30–150 nm) require precise manipulation strategies. Acoustofluidic devices can be tailored to selectively manipulate particles of specific sizes, enabling precise separation and sorting based on size. However, smaller particles, such as nanoparticles (10–1000 nm) used in biomedical applications (e.g., drugs delivery), exhibit different behaviors, with enhanced Brownian motion, which may require different manipulation strategies or higher acoustic intensities to achieve the same manipulation efficiency [16,26]. 

Recent advances in SAW-based micro/nano-particle manipulation have significantly improved the precision and efficiency of acoustic techniques for controlling particles at the nanoscale. The development of high-frequency SAWs and the use of tilted angle SAWs (taSAWs) have enabled more effective manipulation of smaller particles, such as nanoparticles and exosomes [26]. These innovations allow for improved sorting, trapping, and concentration of particles in microfluidic channels with lower energy consumption and greater spatial resolution. Despite these advances, several challenges remain. Manipulating nanoparticles in complex fluids can still lead to aggregation or inefficient handling, and scaling SAW devices for high-throughput applications while maintaining consistent performance is a key issue. Achieving selective and precise manipulation of diverse particle types in real-world applications also continues to present significant hurdles.

There are several literature reviews addressing acoustics for particle/cell manipulation in microfluidics [1,2,7,22,23,24,25,27,28,29,30] that focus on the type of manipulation or final application. However, to the best of our knowledge, this is the first comprehensive, critical, and detailed review of the SAW-based devices developed to manipulate micro- and nano-targets within microfluidic domains (either microchannels or chambers). In this work, we compiled a standardized overview of 43 studies from the last decade, highlighting the type of SAW, channel, and acoustic chip characteristics, and their application in micro- and nanoscales. The next section focuses on the working principles of SAW sensors. Section 3 details how SAWs specifically interact within fluids, elucidating their behavior and effects within microfluidic channels. Additionally, Section 4 features a summary table encompassing numerous studies employing SAWs as actuators to manipulate suspended micro- and nanoparticles. Section 5 provides a comprehensive critical analysis of the various devices identified and their achievements. Finally, we present the main conclusions in Section 6 by highlighting key points and areas for improvement.

## 2. SAW Operating Principles

This section provides a brief overview of the physics underlying SAW sensors and actuators and their fundamental components. We also discuss the parameters and configurations that influence the device’s operation mode and we delve into how SAWs interact with particles in microfluidics, enabling their precise manipulation. As an example of this operation principle, Figure 1 shows a SAW-based microfluidic device for cell sorting.

SAW technology involves a piezoelectric substrate with interdigitated transducers (IDTs) deposited on its surface. When an alternating signal is applied to the electrodes of the IDTs, it generates an oscillating electric field. This field causes the piezoelectric substrate to deform, producing acoustic waves on its surface. The material chosen for the substrate should possess high electromechanical coupling, allowing it to convert mechanical deformation into electrical charge and vice versa [31]. Anisotropic materials play a crucial role in this process due to their directional-dependent properties.

Anisotropic materials exhibit different physical properties, such as speed of sound and elastic constants, depending on the direction of measurement relative to their crystallographic axes. This directional dependence is essential in SAW technology because it influences the propagation characteristics of the acoustic waves. Crystals like quartz, lithium niobate (LiNbO_3_), and lithium tantalate (LiTaO_3_) are examples of anisotropic materials commonly used in SAW devices. Their crystal structure allows for precise control over the propagation of acoustic waves, enabling efficient operation and precise sensing capabilities in applications such as sensors and actuators [23,32]. For instance, LiNbO_3_ with the specific orientation of 128° Y-rotated, X-propagating cut is commonly used in SAW devices because its anisotropic properties facilitate efficient SAW generation and propagation, especially for Rayleigh Waves. According to the acoustic vibration modes and boundary conditions, SAW devices can generate different types of elastic waves such as Rayleigh Waves, Lamb Waves, Love Waves, and Shear Horizontal Waves [32,33]. When the SAW sensor is designed for actuation purposes (e.g., focusing, sorting, and separation) in liquids, Rayleigh Waves are the most commonly used [34,35,36]. Rayleigh Waves are characterized by particle motion that combines both longitudinal and transverse components, resulting in an elliptical motion that decreases with the depth of the substrate. When these waves encounter liquid, they refract at the Rayleigh angle on the surface of the substrate, leading to leaky SAWs. This manifests as acoustic pressure in the liquid, inducing acoustic streaming, pivotal in microfluidic devices [20,32]. 

When designing an acoustic chip, it is essential to consider its intended application and target object. The choice of IDT materials and configurations significantly impacts the propagation of SAWs and, consequently, the performance of the device. Key parameters to consider include wave propagation velocity, coupling coefficient, substrate thickness, finger width and the spacing between them, number of finger pairs, and total aperture length [24]. To better understand the transducer features, Figure 2 shows a schematic representation of a single-electrode IDT with uniform distribution, characterized by equal finger width and spacing, typically corresponding to a quarter wavelength (*λ*/4). 

The propagation velocity and coupling coefficient are dependent on the selected material. The IDT configuration, specifically the finger width and spacing, determine the period of the wave and, consequently, the wavelength of the SAWs (*λ*), which inversely relates to the frequency. Equation (1) shows the relation between these physical quantities:(1)$$\lambda =\frac{v}{f}$$
where v (m/s) is the speed of sound in the piezoelectric substrate and f (Hz) is the frequency of the SAW [22,23]. Hence, a wider finger implies a longer wavelength, leading to a correspondingly lower working frequency.

In these transducers, minimizing reflection is essential for efficient energy transfer and signal transmission [31]. Reflections occur when there is a mismatch between the impedance of the IDT and the medium through which the acoustic wave is propagating. By increasing the spacing between fingers, the impedance mismatch is reduced, resulting in fewer reflections. Furthermore, as more fingers are added to the IDT, the number of sources of acoustic waves increases. This results in a greater number of wavefronts interfering constructively, leading to an overall increase in the amplitude of the acoustic signal produced by the transducer [31,33,37].

Regarding the IDT design, the configurations extend beyond the single-electrode IDT represented in Figure 2. Pursuing precise control in different contexts and targets, researchers have explored alternative designs, including floating electrode, chirped, slanted, and focused IDT configurations, as represented in Figure 3 [38]. 

Floating electrode configuration:

This configuration is used in unidirectional transducers, as unidirectionality is an effective way to suppress insertion loss. The design is similar to the single-electrode design but includes additional electrodes, known as floating electrodes (see Figure 3b). Unlike the main electrodes, these floating electrodes are not connected to any electrical potential and do not contribute to the wave generation. Instead, their purpose is to minimize insertion losses by lowering electrode resistance, reducing parasitic capacitance, improving wave propagation, and providing better impedance matching. These advantages enable higher frequencies within the device, thereby enhancing accuracy. Additionally, they offer lower electrode resistance due to the wider electrodes and enable more effective excitation of higher harmonics, thanks to the increased metallization ratio [24,31,33,39].

Chirped IDTs:

The design of a chirped IDT involves varying the width and spacing of the interdigitated fingers, as shown in Figure 3c, to create a gradient in the frequency response. This gradient allows for the generation of multiple resonance frequencies and enables precise control over propagation properties. Chirped IDTs offer several advantages, including the ability to compensate for dispersion effects caused by substrate materials or device geometry. However, this design can pose challenges during fabrication, requiring sophisticated lithography techniques and precise process control parameters due to its complexity. By tailoring the chirp pattern, these IDTs can achieve customized dispersion profiles tailored to specific application requirements [24,31,32,33]. These types of IDTs, as well as slanted IDTs, are particularly effective for manipulating single particles in stagnant fluid, as varying the frequency allows for precise control of the position of the acoustic pressure nodes, enabling accurate movement of the particle [40].

Slanted IDTs:

Also known as tapered IDTs, in this configuration, the input signal can be tuned across a range of frequencies, similar to chirped IDTs. Tapered IDTs feature a gradual decrease or increase in the spacing between the fingers on one side of the transducer, resulting in a tapered shape (Figure 3d). This design may require complex optimization and fabrication processes. By carefully controlling the angle of the IDT fingers and optimizing the slant geometry, enhanced wave propagation characteristics can be achieved. These enhancements may include increased wave velocity, reduced dispersion, and an improved signal-to-noise ratio [23,33].

Focused IDTs:

Unlike previous designs, focused IDTs, represented in Figure 3e, utilize curved electrodes to concentrate acoustic energy into a narrow beam, pinpointing a small focal point. This precise control over the distribution of acoustic energy enables focused IDTs to attain higher resolution and sensitivity compared to uniform IDTs. Consequently, this configuration proves ideal for SAW sensors and precisely manipulating particles within microfluidic systems [1,2,24].

Table 1 summarizes the key characteristics and applications of various IDT designs discussed, providing a clear overview of their main features and practical uses in SAW-based systems.

Innovations in the design of electrodes for SAW-based microfluidic devices are driven by a common goal: to optimize acoustic wave generation while minimizing signal interference and distortion. Each variant of the IDT presents distinct advantages and challenges. Consequently, the selection of an IDT type is contingent upon the specific performance criteria and operational needs of the microfluidic device. 

When functioning as actuators, SAW devices commonly operate in two distinct modes: traveling surface acoustic wave (TSAW) or standing surface acoustic wave (SSAW), as schematized in Figure 4a,b, respectively [7,20,22]. In the former scenario, a traveling wave is generated on the substrate surface when a single IDT receives an electric signal. On the other hand, in SSAW, two facing IDTs must receive identical input signals to produce two traveling waves propagating in opposite directions. This arrangement results in the formation of a standing field, characterized by the presence of pressure nodes, related to points or regions where the pressure amplitude is minimal, and antinodes, corresponding to points where the amplitude is maximal.

## 3. Acoustic Actuation in Microfluidics

Over the past decade, SAW technology has experienced a significant increase in its application for manipulating fluids and suspended particles within microchannels or chambers. Particularly notable is its role in sorting and separating micro- and nanoparticles, where SAW demonstrates various advantages. These systems provide high precision without the need for labeling and ensure accurate manipulation while preserving particle integrity. Non-contact operation prevents physical damage, and high throughput rates enable efficient processing of large sample volumes. The versatility of SAW technology allows for sorting based on multiple criteria, and low power consumption contributes to energy efficiency. Moreover, SAW-based systems hold potential for mass production, making them promising tools for various applications in research and industry [2]. 

Figure 5 presents two real SAW-based microfluidic devices with different configurations and their corresponding effects on microbeads. These images offer visual insight into the performance of SAW actuators in controlling the positioning and movement of microbeads, demonstrating how different arrangements can be achieved depending on the design and acoustic wave configuration.

A typical acoustofluidic device comprises the aforementioned SAW chip (see Figure 2b), which features metal electrodes, the transducer, deposited on top of a piezoelectric substrate. Additionally, a microfluidic channel is positioned, so that the fluid flows perpendicular to the propagation of the SAWs. In addition to the features of the SAW chip, the material dimensions and geometry of the microchannel also significantly influence the performance of the acoustofluidic platform.

The material most commonly used for fabricating microfluidic channels in SAW-based devices is polydimethylsiloxane (PDMS) [14,42,43,44,45,46], but channels fabricated with other materials can also be found in the literature, such as polymethyl methacrylate (PMMA) [47], glass [48], and silicon [34]. PDMS presents advantages, such as ease of fabrication, flexibility, transparency, and biocompatibility. PMMA offers similar advantages, including improved optical properties, but it is not porous, is more rigid, and may be more prone to mechanical failure. Glass microchannels are used because of their optical transparency and compatibility with various detection methods. On the other hand, silicon provides superior mechanical and chemical properties but requires more complex and expensive fabrication processes.

In terms of dimensions, it is crucial to align the width of the channel with the wavelength of the SAW, particularly in SSAW systems, as this alignment determines the number of pressure nodes within the channel. For instance, if the width is close to half of the wavelength, there will be one pressure node positioned at the center of the channel [23,49,50]. Unlike passive microfluidic techniques, the operation of SAWs does not need complex or sophisticated channel designs. Instead, a straightforward linear configuration with multiple inlets and outlets is commonly employed [17,42,44,51]. The incorporation of multiple inlet holes often corresponds to the implementation of sheath flows. This microfluidic technique involves the controlled circulation of a surrounding fluid, known as the sheath fluid, around a central sample stream. Such configuration serves various purposes, including sample concentration, alignment, sorting, and preservation by avoiding the contact with the channel walls [52,53].

By adjusting these parameters within the microfluidics domain, to match the characteristic sizes of the particles of interest, the efficiency, resolution, and throughput of microfluidic sorting and separation techniques can be enhanced. To gain a better understanding of how the entire system interacts, it is essential to explore the forces involved in both SSAW and TSAW technologies. 

### Governing Equations

The governing equations that are typically used to describe the behavior of SAW-based microfluidic devices encompass the fundamental principles of piezoelectricity, acoustic wave propagation, and fluid dynamics within the context of microfluidics.

Firstly, piezoelectricity is the foundation for generating SAWs. The constitutive equations of piezoelectricity, which relate to the mechanical stress, strain, electric field, and electric displacement, are crucial for understanding this process. The tensor stress and the electric displacement equations (Equation (2) and Equation (3), respectively), considering linear piezoelectricity, are typically expressed as [40,54](2)$$T_{ij}=c_{ijkl}^{E}S_{kl}-e_{kij}E_{k}$$(3)$$D_{i}=e_{ikl}S_{kl}+\varepsilon_{ik}^{S}E_{k}$$
where $c_{ijkl}^{E}$ is the stiffness tensor at constant electric field, $S_{kl}$ and $e_{kij}$ are the components of strain tensor and the stress tensor, respectively, $E_{k}$ is the electric field and $\varepsilon_{ik}^{S}$ is the permittivity tensor at constant strain.

Secondly, acoustic wave propagation describes how the generated SAWs travel along the surface of the piezoelectric material and interact with the fluid in the microfluidic channel [55,56]. In such a multiphysics problem, the amplitude of the acoustic pressure in the Helmholtz wave equation results from the pressure outputted by the piezoelectric problem equations. Flow and particle behavior play a crucial role in understanding the mechanisms used to manipulate cells in microfluidic devices. The distribution of acoustic pressure generated by surface acoustic waves in water can be considered a harmonic problem governed by the Helmholtz wave equation [43,50,51]:(4)$$\nabla^{2}p_{a}-\frac{\omega^{2}}{u_{a}^{2}}p_{a}=0$$
where $p_{a}$ is the amplitude of the acoustic pressure, *ω* is the angular frequency and u is the speed of the sound in the medium.

Lastly, fluid dynamics within microfluidics involves understanding the behavior of the fluid under the influence of the acoustic waves (namely their acoustic pressure fields). The Navier–Stokes equation in Newtonian fluids (Equation (5)), along with the continuity equation (Equation (6)), are essential to describe the fluid motion. These equations account for the conservation of momentum and mass, respectively [57]:(5)$$\rho \left[\frac{𝜕v}{𝜕t}+\left(v\cdot \nabla \right)v\right]=-\nabla p+\mu \nabla^{2}v+f$$
and(6)$$\frac{𝜕\rho }{𝜕t}+\nabla \cdot \left(\rho v\right)=0$$
which can be simplified for incompressible fluids as(7)∇⋅v=0

In Equations (5)–(7), *ρ* is the fluid density, v is the fluid velocity, *p* is the pressure, *μ* is the dynamic viscosity, and *f* represents body forces, such as those due to acoustic streaming.

When particles are suspended in a fluid and exposed to SSAWs, they undergo complex interactions with the surrounding medium, experiencing two primary forces: drag force (*F_d_*) induced by acoustic streaming flow, and acoustic radiation force (*F_R_*). The calculation of *F_d_* (Equation (2)) derives from Stokes law, which describes the drag force experienced by small spherical particles moving through a viscous fluid at low Reynolds numbers [23,42].(8)$$F_{d}=6\pi \mu Rv$$
where *μ* is the shear viscosity of the fluid (Pa·s), *R* is the radius of the spherical particle (m), and *v* is velocity of the particle relative to the fluid (m/s). The magnitude of the acoustic radiation force in a standing wave field, acting on spheroids, is given by [23,58]:(9)$$F_{R}=-\left(\frac{\pi P_{0}^{2}V_{p}\beta_{f}}{2\lambda }\right)\varphi \left(\beta ,\rho \right)\;sin(2kx)$$(10)$$\varphi \left(\beta ,\rho \right)=\frac{5\rho_{p}-2\rho_{f}}{2\rho_{p}+\rho_{f}}-\frac{\beta_{p}}{\beta_{f}}$$
where *P*_0_ is the acoustic pressure (Pa), *V_p_* is the particle volume (m^3^), *β_f_* and *β_p_* are the compressibility of the fluid and of the particle (Pa^−1^), respectively, *ρ_f_* and *ρ_p_* are the density of the fluid and of the particle (Kg/m^3^), respectively, *λ* is the wavelength (m), *k* the wavenumber (m^−1^), *x* the distance to a pressure node (m), and φ the acoustic contrast factor.

To illustrate the forces involved in the operation of the SSAW microfluidic system, a schematic cross-sectional view is provided in Figure 6, showing the key interactions and pressure distributions.

Upon analysis of Equations (9) and (10), it becomes evident that the contrast factor is influenced by both the density and compressibility of the particle and the surrounding medium. However, the significance of the contrast factor lies in its role in determining the direction of *F_R_* and subsequent particle movement. A positive contrast factor results in attraction towards pressure nodes, while a negative contrast factor leads to movement towards antinodes [2,3,26]. Most cells and particles exhibit positive acoustic contrast factors in water-based solutions, as detailed in [60].

Additionally, the particle size relative to the fluid wavelength determines the dominant force. When the particle perimeter is smaller than the fluid wavelength, it primarily experiences the drag force, induced by acoustic streaming flow. On the other hand, if the particle perimeter exceeds the fluid wavelength, the acoustic radiation force becomes dominant, primarily pushing the particles. As particles diminish in diameter to nanometer dimensions, the decline in acoustic radiation force outpaces that of the hydrodynamic viscous force. This fact presents a challenge for SAWs in precisely manipulating nanoparticles. Thus, the influence of each force component on the particles depends on their size, density, and compressibility, highlighting the complex interplay between particle characteristics and the acoustic environment [2,22].

Traveling acoustic waves can also manipulate particles in microfluidic systems through similar forces. However, unlike standing waves, these waves deflect particle trajectories without forming a distinct pressure node pattern field. TSAWs exert a force on suspended particles through momentum transfer from the wave to the particles. As the traveling wave propagates through the fluid medium, it imparts kinetic energy to the particles, causing them to experience a force that deflects their trajectories. This force arises from the interaction between the acoustic wave and the particles, leading to changes in particle motion [25,30]. The magnitude and direction of this force depend on various factors, including the properties of the particles, such as size and density, as well as the characteristics of the wave, such as frequency and amplitude. The main applications of TSAWs as actuators are microfluidic mixing, concentration, patterning, and transport [30,61,62].

While both TSAWs and SSAWs can manipulate particles using acoustic streaming and acoustic radiation force, the specific mechanisms and outcomes may vary based on the wave type and the design of the microfluidic device. Understanding these intricate interactions between particle characteristics and the acoustic environment is crucial for optimizing SAW-based microfluidic systems and advancing nanoparticle manipulation techniques. 

## 4. SAW-Based Microfluidic Devices for Manipulation

Extensive research was conducted to gather and compare SAW-based microfluidic platforms, primarily aimed at micro and nano bioparticle focusing, sorting, and separation tasks. The research parameters encompass articles published since 2011, focusing on the keywords ‘surface acoustic waves’, ‘microfluidics’, and ‘particle/cell manipulation’. In this section we summarize the works reported in the literature that consider only SAW-based microfluidic devices for the manipulation of particles and cells. Studies utilizing different techniques were excluded here. Also, studies lacking experimental results were not included.

Table 2 presents a concise summary of devices and their experimental results sourced from scientific databases, aligning with a goal of exploring SAW technology for actuation in microfluidic systems. The selected studies were categorized based on operational mode, beginning with those utilizing TSAWs, followed by devices employing both TSAWs and SSAWs, and concluding with those using solely SSAWs. The collected information includes the IDT design and features, SAW operation wavelength, microfluidic channel characteristics, flow rates, device purpose, type of target and dimensions, as well as most relevant reported results. In most cases, the data were presented keeping the authors’ terminology. The structured resume table enhances knowledge extraction, providing a streamlined approach to compare different methods, applications, and results.

While Table 2 focuses on SAW-based microfluidic devices for particle manipulation, it is important to highlight that these same principles are equally relevant to sensing applications, although the operating frequencies for sensing applications are typically higher, in the hundreds of megahertz to gigahertz range. Particle behavior under acoustic forces can be utilized to detect changes in fluid properties or to analyze biological samples based on size, shape, and mechanical properties. Additionally, SAW-driven manipulation techniques have the potential to enhance sensor sensitivity and selectivity by improving target localization and interaction efficiency at the sensor surface.

## 5. Discussion 

To conduct this literature review, each study was thoroughly examined, enabling us to gather insights beyond what is presented in Table 2. Consequently, this section will delve into both the key findings included in Section 4 and those omitted, as well as elucidate common trends observed across the studies.

### 5.1. Materials and Fabrication

Although the literature reports different materials adequate for acoustic actuation applications, notably, all the devices employed the same piezoelectric substrate: a wafer of LiNbO_3_ with a crystal cut orientation of 128° Y-rotated, X-propagating. The materials used for the IDTs vary, with chromium (Cr), gold (Au), aluminum (Al), titanium (Ti), and silver (Pt) being the most common materials. Furthermore, a Cr/Au mixture, with ratios of 1:10 and 2:8, is particularly prevalent. The reported fabrication processes were very similar across the majority of the works, both for the SAW transducer and the channel. The typical fabrication process involves the patterning and deposition of the IDT materials onto the substrate, using photolithography [35,87] and e-beam evaporation techniques [14,64], followed by a lift-off process [15,79,88]. Regarding IDT design, the predominant configuration was the single-electrode design with one-quarter SAW wavelength finger width and spacing [37,67,84,86]. This configuration is favored due to its simplicity in fabrication, and it shows good results with targets at the microscale.

### 5.2. Microfluidic Domain

The microfluidic components consistently use PDMS as the primary material for channel fabrication, employing a standard soft-lithography and mold-replica procedure, as described in [89]. This preference is justified by PDMS chemical and mechanical properties, such as optical and acoustic transparency, which minimize energy loss, flexibility enabling complex channel designs, biocompatibility, and suitability for the fast development of prototypes [5]. Additionally, Zhao et al. [14] employed a different approach by incorporating a mixture of soft and hard PDMS to enhance the acoustic transmission to the microfluidic domain. Numerical simulations demonstrate that the hybrid hard/soft PDMS channel exhibits a substantial increase in acoustic pressure transmission, rising from 61.9% in the standard soft PDMS channel to 92.1% [14]. Mao et al. [34] investigated the performance of silicon and PDMS narrow channels in cell separation. To examine the wall effect, they employed these two materials to induce different contrasts in acoustic impedance between the channel and the fluid. This contrast is higher when using hard materials, such as silicon, when compared to PDMS. In another study, the same authors chose glass to fabricate a square capillary, in which nanoparticles were manipulated in a static sample [68]. 

Channels with two or three inlets for sample and sheath flow, along with two or three outlets corresponding to the range of particle sizes for separation, were commonly found in the studies reported in the literature (see Figure 7a). Variations in channel dimensions, specifically height and width, were observed across different studies. Although several authors explored multiple configurations, a predominant trend emerged with channels exhibiting a greater width-to-height ratio. No consistent pattern appeared in the flow rates, as a wide range of values was encountered, from 0.04 μL/min to 50 μL/min and from 0.33 mm/s to 83.3 mm/s. However, in most devices, the flow rate in the sheath flow inlets is significantly higher than that of the sample inlet, which requires a lower velocity, so the micro- or nanoparticles can be effectively actuated by the acoustic waves. 

From the reported works, it is observed that the typical device consists of a microfluidic channel bonded to a substrate to form a single platform. However, some authors [67,81] have developed a detachable acoustofluidic platform that allows for the replacement of the channel while reusing the SAW chip. This feature could be particularly valuable in medical and chemical applications where sample preservation from cross-contamination is paramount. 

### 5.3. SAW Operation Modes

Out of the 43 studies analyzed in Table 2, 24 employed the SSAW operational mode. This included variations such as tilted-angle SSAW (taSSAW) in seven works, phase-shifted SSAW in five works, SSAW Lamb waves instead of Rayleigh waves in two works, and a cascade SSAW in one. In contrast, TSAW was utilized in 17 studies, with one of them employing tilted-angle TSAW (taTSAW). Two studies applied both modes, with one noteworthy aspect being that the SAW chip employed a pair of IDTs to generate SSAWs, while TSAWs were generated by a different IDT (see Figure 7b). 

In the usual configuration, wave propagation is perpendicular to the fluid flow in the channel. However, in taTSAW or taSSAW devices, the IDTs are patterned at a slight angle relative to the flow direction, as shown in Figure 7c. This technique, commonly employed in focusing and separation devices, entails tilting the IDTs typically between 5 and 15 degrees to enhance sensitivity and efficiency. By doing so, the acoustic waves generate inclined pressure nodal lines, thereby increasing the migration distance of suspended particles [5,11,12,23]. Moreover, research has demonstrated that tilted-angle devices can mitigate the effects of the anechoic corner issue [1,29,42]. This phenomenon typically occurs near the channel inlet, along the channel wall, where the streaming is strong and wave reflection causes low acoustic force. Consequently, some particles may deviate from their intended path, being pushed towards the channel wall instead of aligning with the pressure nodes and antinodes, compromising the effectiveness of particle sorting and separation processes.

Phase shift SSAW technology manipulates pressure node locations in a microfluidic channel by adjusting the phase difference of the input signal between the pairs of IDTs. This facilitates precise control over particle or cell positions, finding applications in three-dimensional (3D) trapping, particle sorting, and separation. Similarly to taSSAW, this method demonstrates the ability to enable particle translation over extended distances, potentially surpassing the limitations imposed by traditional SSAW systems, which are typically limited to a quarter of the SAW wavelength [50,77].

The approach by Ng and Neild [82] involved successive pairs of IDTs positioned along the length of the channel, operating independently. This configuration resulted in pressure nodes with a slight horizontal offset, generating multiple particle trajectories. With the cascade SSAW method, the authors successfully sorted 5 µm particles into four distinct outlets using different actuation combinations. 

### 5.4. Working Ranges

The wavelengths for TSAW devices have consistently ranged in the tens of micrometers, whereas in SSAW studies, they typically fall within the hundreds of micrometers, decreasing the working signal frequencies and simplifying the electronics involved. Consequently, the operation frequencies of SAWs in TSAW devices are typically in the hundreds of megahertz range, whereas in SSAW studies, they operate at tens of megahertz. 

### 5.5. Applications

The most common application of SAW-based actuators lies in the separation of bioparticles by size, particularly on the microscale. Nevertheless, in [51] and later in [83], the authors focused on separation by density, a key approach when the targets have very similar sizes. 

The works by Nguyen et al. [80] and Guo et al. [59] (see Figure 7d) stand out for their 3D cell manipulation capabilities, contrasting with the predominant horizontal control in most devices. Both studies focused on precisely controlling single or groups of cells within a closed-loop environment, conducting experiments with static samples rather than continuous flow.

To achieve this, they designed devices with two sets of perpendicular IDTs and utilized SSAW fields and phase shifting to relocate cells to specific coordinates. 

Regardless of their ultimate research objectives, over 90% of the studies in Table 2 initially tested with polystyrene (PS) beads. These beads are often chosen for their standardized size, shape, and availability, providing a consistent model particle for testing and proof of concept. Their ease of handling within microfluidic systems simplifies experimental setup and manipulation procedures. Furthermore, results obtained with PS beads serve as a valuable reference point for comparing the performance of different microfluidic devices and experimental techniques. By validating microfluidic platforms with PS beads before transitioning to more complex biological samples, researchers ensure the reliability and reproducibility of their results. Studies reported in the literature [14,43,78] demonstrate a good agreement between the results obtained using particles and cells.

### 5.6. Accuracy and Efficiency

Upon comparing the results presented in this review, it can be observed that there is no clear consensus among researchers regarding the SAW operation method that offers better accuracy and efficiency, nor the optimal IDT design or configuration. We observed that IDTs with more complex designs, such as chirped, slanted (see Figure 7e), and focused, are frequently used to generate more efficient traveling waves, which offers very good results in focusing and sorting micro- and even nanoparticles, as shown in [16,66,68]. The simpler single-electrode design is widely employed to generate standing acoustic fields, where the studies are focused on achieving a desired pressure node and antinodes pattern, through variations of the SAW operating mode, including tilted-angle and phase shift. 

While TSAWs are primarily utilized for mixing, sorting, and separation, SSAWs serve a broader range of purposes. In addition to the applications shared with TSAWs, SSAWs also offer precision trapping and even enable three-dimensional manipulation. 

Comparing quantitative accuracy and purity separation results between studies is not feasible, due to the numerous variables involved and the lack of uniformity in the results’ presentation by the authors. However, it is noteworthy that the majority of devices with available quantitative results demonstrate successful values, exceeding 90% efficiency. 

### 5.7. Other Considerations

While not included in this review, numerous numerical simulation works have been reported, and are highly relevant in this field [49,90,91,92,93]. These works present numerical models (for instance, finite element or finite volume based) and their simulations to predict the behavior of waves, fluids, and suspended particles, enabling the testing of various configurations without the associated costs and efforts of fabrication and experimentation. As a result of these previous numerical approaches, many experimental studies begin by presenting corresponding simulations, which often closely align. This practice contributes to optimizing devices and significantly advances the state of the art in SAW-based microfluidic devices.

## 6. Conclusions and Future Perspectives

The field of SAW-based microfluidics has experienced notable progress in recent years, driven by the unique capabilities of SAW technology in manipulating particles and cells at the micro- and nanoscales. This advancement has been further enabled by the evolution of microfabrication techniques. Through an extensive literature review, we have delved into various designs, configurations, and methodologies employed in SAW-based microfluidic devices. Standardizing all the data has facilitated the extraction of valuable insights into trends in materials, fabrication techniques, as well as IDT and channel designs and configurations. 

However, our review revealed a lack of consensus regarding the optimal SAW technique and device settings for manipulating micro- and nanoparticles within a microfluidic channel. Despite this, we have identified critical commonalities that serve as valuable starting points for future research endeavors.

Moreover, comparing techniques and analyzing their advantages and limitations will prove instrumental in directing future research efforts. This will enable the exploration of novel applications and facilitate the translation of laboratory findings into practical solutions for real-world challenges in healthcare, environment, and biotechnology sectors. 

Looking towards the future, SAW devices are poised to be highly promising candidates for lab-on-chip applications across various fields, including clinical diagnosis, laboratory research, food safety, and environmental monitoring. This technology is anticipated to enable precise, high throughput, and continuous manipulation of small targets and multimodal operations in point-of-care applications [94]. For instance, Husseini et al. demonstrated the potential to create portable point-of-care DNA testing, allowing for rapid and accurate DNA testing in diverse settings [95].

Additionally, the ability to manipulate living organisms without contact is gaining traction, proving significant for tasks such as single-cell analysis and imaging motile organism morphology. This capability is particularly beneficial for tissue engineering-related projects, where maintaining cell integrity is crucial [96]. Furthermore, research has highlighted that living cells respond differently to acoustofluidic stimulation depending on their physiological states, which can be useful in immunotherapy applications or even for fertility treatments [97,98].

It would be particularly remarkable to explore the development of an acoustofluidic platform that integrates multiple applications currently performed separately. For example, utilizing SAW technology to manipulate and sense cells within a single microfluidic platform. Moreover, the future of SAW-based microfluidics lies in its convergence with other technologies such as magnetic, optical, chemical, and artificial intelligence methods, fostering new multidisciplinary technologies and therapeutic strategies. Nevertheless, challenges remain, particularly in scaling up the manufacturing of these devices. Advancements in manufacturing techniques, such as 3D printing and soft lithography, could reduce the production costs of SAW-based microfluidic devices. This would make the technology more accessible for widespread use in research and clinical settings. Research into alternative, cost-effective materials for substrates and IDTs could further lower production expenses without compromising device performance.

## Figures and Tables

**Figure 1 sensors-25-01577-f001:**
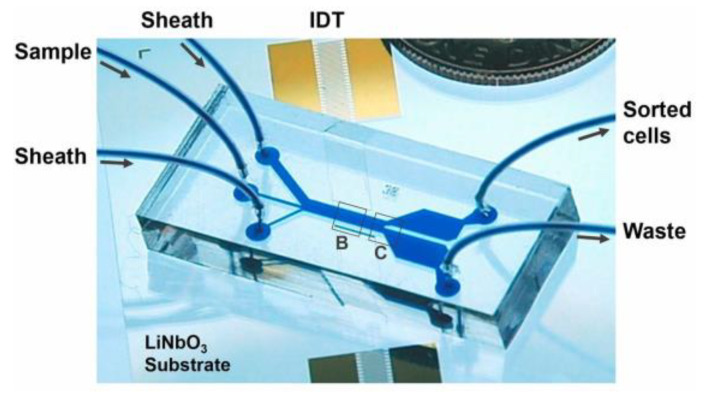
Example of a SAW-based microfluidic device for cell sorting. Adapted from [11], PNAS, under a Creative Commons Attribution (CC BY) license.

**Figure 2 sensors-25-01577-f002:**
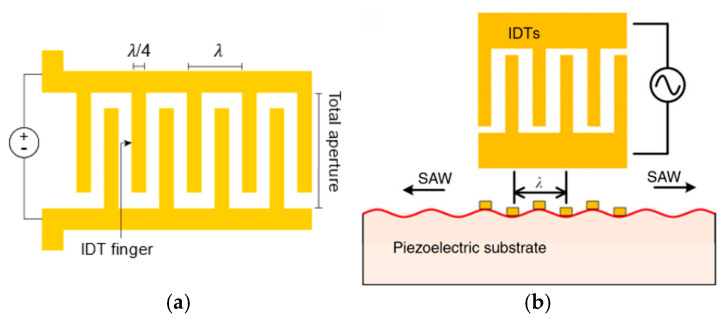
(**a**) Schematics of an interdigitated transducer (IDT), showing the relation between the finger pairs and the wavelength of the resulting surface acoustic wave (SAW). (**b**) A typical SAW device, with the IDT patterned on the surface of the piezoelectric substrate. Reproduced from [30] with permission from Nature.

**Figure 3 sensors-25-01577-f003:**
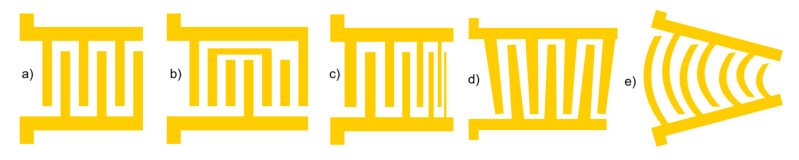
Scheme of different IDT designs: (**a**) single electrode; (**b**) floating electrode; (**c**) chirped; (**d**) slanted; (**e**) focused.

**Figure 4 sensors-25-01577-f004:**
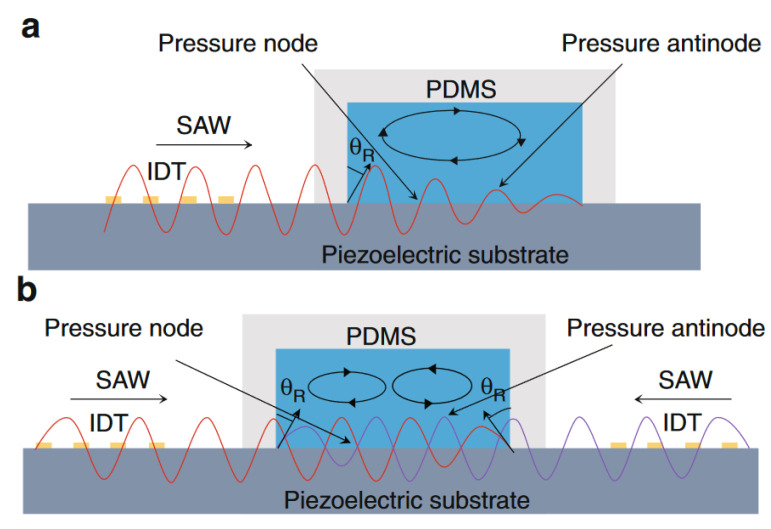
Schematic representation of a cross-sectional view of the two operation modes of SAW microfluidic devices. (**a**) Traveling SAW operation mode. (**b**) Standing SAW operation mode. Adapted from [23] with permission from Nature.

**Figure 5 sensors-25-01577-f005:**
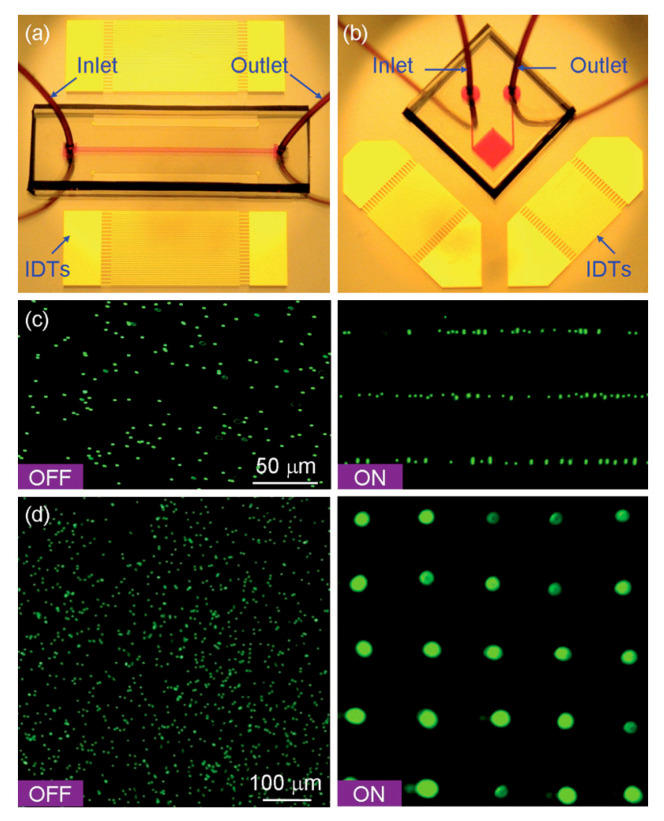
Manipulation of fluorescent polystyrene beads. (**a**) 1D manipulation; (**b**) 2D manipulation; (**c**) microbead experimental patterning (1D); (**d**) microbead experimental patterning (2D). Reprinted with permission from [41] Copyright 2009, Royal Society of Chemistry.

**Figure 6 sensors-25-01577-f006:**
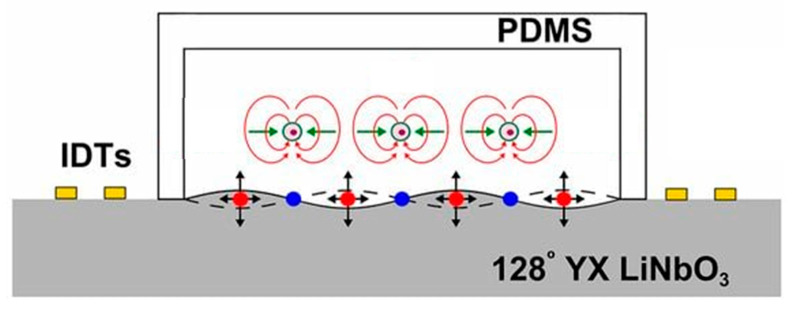
Schematic of a cross-sectional view of the SSAW microfluidic device, highlighting the forces acting within the system. The substrate generates Rayleigh waves, which induce both longitudinal and transverse motions. These motions give rise to acoustic streaming (depicted by red arrows) and the acoustic radiation force ($F_{R}$ represented by green arrows). Blue dots indicate the pressure nodes, where the acoustic pressure amplitude is minimal, while red dots represent the pressure antinodes, where the acoustic pressure amplitude is maximal. Adapted from [59], PNAS, under a Creative Commons Attribution (CC BY) license.

**Figure 7 sensors-25-01577-f007:**
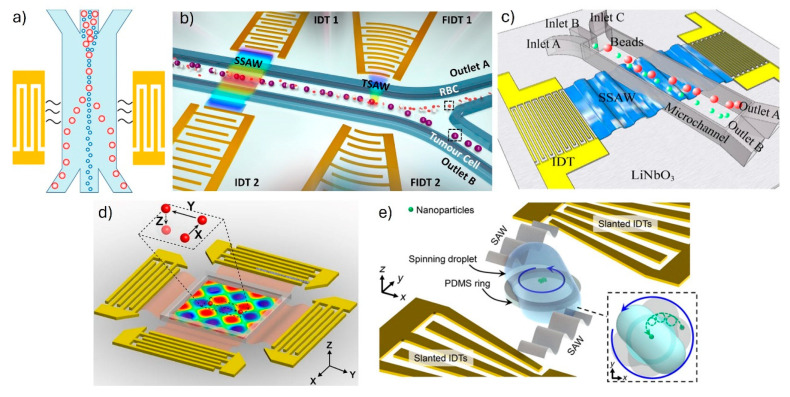
Schematic representation of different configurations of SAW-based microfluidic devices. (**a**) A SSAW-based device, for separation, with single electrode IDTs and a microfluidic channel with 3 inlets and 3 outlets. Reprinted from [13], American Chemical Society, under a Creative Commons Attribution (CC BY) license. (**b**) A device using SSAWs for concentration and TSAWs for separation. TSAWs are generated alternately by the focused IDTs on both sides of the channel. Adapted from [10], with permission from Elsevier. (**c**) A tilted-angle SSAW device for separation. Adapted from [42], American Chemical Society, under a Creative Commons Attribution (CC BY) license. (**d**) A SSAW-based device for 3D manipulation with two pairs of single-electrode IDTs arranged orthogonally to each other. Adapted from [59], PNAS, under a Creative Commons Attribution (CC BY) license. (**e**) A device with two slanted IDTs to generate TSAWs and manipulate a static sample. Adapted from [16], American Association for the Advancement of Science, under a Creative Commons Attribution (CC BY) license.

**Table 1 sensors-25-01577-t001:** Summary of key features and applications of different IDT designs.

IDT Type	Key Characteristics	Applications
Floating electrode	Incorporates additional unconnected electrodes to reduce insertion loss and enhance unidirectional wave propagation.	High-frequency SAW devices, improved impedance matching, enhanced signal clarity.
Chirped	Varies finger spacing for a broad frequency response.	Frequency-tunable SAW devices, precise manipulation, broadband signal processing, acoustic tweezer modulation.
Slanted	Gradually decreases finger spacing on one side, enabling frequency-dependent SAW excitation.	Multi-frequency operation, frequency filtering, size-selective manipulation.
Focused	Uses curved electrodes to concentrate acoustic energy at a focal point.	High-resolution sensors, precise particle manipulation, acoustic trapping.

**Table 2 sensors-25-01577-t002:** List and detailed characteristics of the reported SAW-based devices.

Team	Year	SAW Type	IDT Design	λ_SAW_ (μm)	Microfluidic Domain	Goal	Target Dimensions	Main Results
Skowronek, Viktor et al. [63]	2013	TSAW	1 slanted IDT Widths = 800 or 1200 or 800 μm Finger pairs = 12 or 65 or 25	[24.7–16.5] or [10.9–9.1] or [5.6–4.6]	Channel with 3 inlets and 2 outlets. Dimensions: height = 50 μm; width = 200 μm; Flow rate: sample = 120 μL/h	Study deflection, sorting	Microscale: 2, 3, 4.5 and 10 μm PS particles	All beads deflected except 2 μm particles, which followed the flow field. The smaller particles deflected at higher frequencies (265–325 MHz).
Destgeer, Ghulam et al. [64]	2014	TSAW	1 focused IDT Finger pairs = 30 Degree of arc = 40° Focal length = 4 mm	20 or 25 or 30	Channel with 3 inlets and 2 outlets. Dimensions: height = 40 μm; width = 200 or 500 μm; Flow rate: sample = 2.5 μL/h; sheath flow = 22.5 and 100 μL/h	Separation	Micro- and nanoscales: 710 nm, 3, 3.2, 3.4, 4.2, 4.5, 5 μm PS particles	Successful separation of PS particles having size differences as low as 200 nm.
Collins, David et al. [65]	2015	TSAW	1 focused IDT Finger pairs = 36 Total proximal aperture = 56 μm Total distal aperture = 210 μm Degree of arc = 26°	10	Channel with 3 inlets and 3 outlets. Dimensions: height = 20 μm; width = 40 or 80 μm; Flow rate: sample = 0.5 or 1 μL/min; sheath flow = 3 and 8 μL/min	Rapid Sorting	Microscale: 1, 2, 3 μm particles	Highly focused SAW enabled deterministic sorting with a narrow beam width of 25 μm, frequency of 386 MHz and λ = 10 μm.
Collins, David et al. [66]	2016	TSAW	1 focused IDT Finger pairs = 42 Terminal aperture = 14 μm Degree of arc = 26°	6 or 10	Channel with 1 inlet and various outlets. Dimensions: height = 20 or 40 μm; width = 160 or 400 μm; Flow rate = 0.45 μL/min	Focusing	Micro- and nanoscales: 1, 2 μm, and 100, 300, 500 nm PS particles	Demonstrated the streaming-induced manipulation of particles with diameter below 1 μm.
Ma, Zhichao et al. [67]	2016	TSAW	1 single-electrode IDT Width = λ/4 Spacing = λ/4	160 or 80 or 40	Channel with 2 inlets and 2 outlets Dimensions: height = 25 μm; width = 400 μm; length = 7 mm Flow rate: sample = 5 μL/min; sheath flow = 8 μL/min	Separation	Microscale: 10 and 15 μm PS particles	Separation efficiency of 98% (for λ = 80 μm).
Mao, Zhangming et al. [68]	2017	TSAW	2 chirped IDTs Spacing = [50–160] or [140–330] μm Finger pairs = 20 Total aperture = 30 mm	≈[202–633] or [570–1209]	Channel with 1 inlet and 1 outlet (static sample) Dimensions: height = 100 or 200 μm; width = 100 or 200 μm; length = 10 mm	Focusing and enrichment	Nanoscale: 80, 200 nm SiO_2_ and 110, 220 and 500 nm PS particles + streptavidin	Successful focusing of particles of all sizes and detection of streptavidin at a concentration as low as 0.9 nM.
Ahmed, Husnain et al. [45]	2018	TSAW	1 single-electrode IDT placed beneath the microchannel Width = 6.5 μm Spacing = 6.5 μm Finger pairs = 20 Total aperture = 0.5 mm	26	Channel with 2 inlets and 2 outlets Dimensions: height = 80 μm; width = 250 μm Flow rate: sample = 50 μL/h; sheath flow = 450 μL/h	Vertical focusing and separation	Microscale: 4.8, 3.2 and 2 μm PS particles	Successful separation of PS particles, presenting purity > 97% and recovery rate > 99%.
Fakhfouri, Armaghan et al. [37]	2018	TSAW	1 single-electrode IDT Width = λ/4 Spacing = λ/4 Total aperture = 750 μm	15 or 21 or 25 or 36	Channel with 1 inlet and 1 outlet (static sample) Dimensions: height = 26 μm (also studied 9.5 and 35 μm); width = 200 μ; length = 13.5 mm	Study patterning	Micro- and nanoscales: 100, 300, 500 nm, and 1, 2, 3, 5 μm PS particles	Observation of the transition in acoustophoretic behavior of particles based on particle diameter, channel height, frequency, and intensity of the TSAW.
Nam, Jeonghun et al. [69]	2018	TSAW	1 focused double-electrode IDT Width = 100 μm Spacing = 100 μm Finger pairs = 20 Degree of arc = 60°	800	Channel with 2 inlets and 1 outlet Dimensions: width = 300 μm Flow rate: 5–70 μL/min	Mixing	NA	With an applied voltage of 21 V, the mixing efficiency was greater than approximately 97% at a flow rate of Q ≤ 80 μL/min.
Ahmed, Husnain et al. [53]	2018	taTSAW	2 single-electrode IDTs placed beneath the microchannel Finger pairs = 20 Total aperture = 0.5 mm Tilted angle = 30°	19 or 26	Channel with 1 inlet and 2 outlets Dimensions: height = 20 μm; width = 500 μm Flow rate = 5.56 mm/s or 83.3 mm/s	Focusing and separation	Microscale: 4.8 and 3.2 μm PS particles	High purity > 99% at both outlets (lower flow rate), one outlet decrease purity to >93% at higher flow rate.
Ahmed, Husnain et al. [70]	2019	TSAW	1 single-electrode IDT placed beneath the microchannel Width = 6.5 μm Spacing = 6.5 μm Finger pairs = 20 Total aperture = 1 mm	26	Channel with 2 inlets and 2 outlets Dimensions: height = 80 μm; width = 250 μm Flow rate = 50–400 μL/min	Mixing	NA	The mixing efficiency reached 100% at 12 Vpp and flow rate of 50 μL/min. As the flow rate was increased to 200 μL/min, the mixing efficiency decreased to 90%.
Lim, Hyunjung et al. [71]	2020	TSAW	1 focused IDT Width = 25 μm Spacing = 25 μm Finger pairs = 80 Degree of arc = 10°	100	Chamber with 2 inlets and 1 outlet Dimensions: diameter = 3 mm Flow rate: 50–450 μL/min	Mixing	NA	The mixing index (with voltage of 20 V) was greater than 0.9 at a total flow rate of Q ≤ 300 μL/min.
Liu, Guojun et al. [44]	2020	TSAW	1 focused IDT Width = 7.5 μm Spacing = 7.5 μm Finger pairs = 35 Degree of arc = 40° Focal length = 4 mm	30	Channel with 3 inlets and 2 outlets Dimensions: height = 50 μm; width = 300 μm Flow rate: sample = 0.33 mm/s; sheath flow = 3.33 and 0.33 mm/s	Sorting and separation	Microscale: 1 and 10 μm PS particles	Separation and sorting efficiency over 99%.
Gu, Yuyang et al. [16]	2021	TSAW	2 slanted IDTs (3 different configurations) Width and spacing decreased from: [140–70] μm [75–35] μm [2.5–17.5] μm	≈[570–285] or [307–143] or [133–71]	Two configurations: 1. Circular open chamber; 2. Circular open chambers connected by a channel. Dimensions: chamber radius proportional to droplets; channel height = 100 μm; channel width = 200 μm	Enrichment and separation	Nanoscale: Droplets + DNA segments and exosome subpopulations	Achieve isolation of different exosome subpopulations with purity > 80%.
Hsu, Jin-Chen and Chang, Chih-Yu [72]	2022	TSAW + APM	2 single-electrode IDTs (1 transducer and 1 receiver) Width = 100 μm Spacing = 100 μm Finger pairs = 20 Total aperture = 1.2 mm	400	Channel (semicircular) with 2 inlets and 2 outlets Dimensions: diameter = 800 μm; length = 6 mm Flow rate: 50–450 μL/min	Mixing	NA	Efficient acoustic mixing for a continuous flow in a semicircular microchannel actuated by SAWs and coupled plate modes.
Cha, Beomseok et al. [73]	2023	TSAW (parallelpropagation to the flow)	1 single-electrode IDT placed beneath the microchannel Width = 6.5 μm Spacing = 6.5 μm Finger pairs = 30 Total aperture = 1.5 mm	26	Channel with 2 inlets and 1 outlet Dimensions: height = 160 μm; width = 600 μm; length = 12 mm Flow rate: sample = 10 μL/min; sheath flow = 40 μL/min. Also tested flow rates between 50–200 μL/min.	Mixing and cellular lysis	Microscale: 6.1 μm PS particles and human RBC	Rapid, controlled flow mixing at low power (< 6.0 Vrms) and high throughput (∼ 0.2 mL/min) with viscous fluids. High lysis efficiency (> 90%).
Khan, Muhammad S. et al. [17]	2024	TSAW	1 slanted IDT Width = λ/4 Spacing = λ/4 Finger pairs = 40 Total aperture = 1 mm	[6.5–8.5] or [5–7]	Channel with 3 inlets and 3 outlets Dimensions: height = 67 μm; width = 500 μm Flow rate: sample = 20 μL/h; sheath flow = 30 and 200 μL/h	Separation by shape	Microscale: 4 and 6 μm microspheres, prolate ellipsoids, and peanut-shaped PS particles/ Thalassiosira eccentrica as bioparticle	High purity and recovery rate of the separated spherical and peanut shaped PS microparticles (percentages details by size in [17]).
Devendran, Citsabehsan et al. [74]	2016	SSAW + TSAW	4 chirped IDTs (Two pairs arranged orthogonally to each other) Width = [20–70] μm Finger pairs = 34 Total aperture = 1140 μm	≈[67–44] or [57–33]	Chamber (static sample) Dimensions: height = 25 μm; width = 707 μm; length = 707 μm	Focusing and separation	Microscale: 3.1, 5.1, and 7 μm PS particles	TSAW component pushed larger particles across the chamber, while smaller particles were collected at the center by SSAW for both mixtures (5.1 and 7 μm, and 5.1 and 3.1 μm).
Wang, Kaiyue et al. [10]	2018	SSAW + TSAW	2 single-electrode IDTs + 2 focused IDTs Width = λ/4 Spacing = λ/4	130 (SSAW) 100 (TSAW)	Channel with 1 inlet and 2 outlets Dimensions: height = 50 μm; width = 65 μm Flow rate = 0.3 μL/min	Focusing and separation	Microscale: 2 and 5 μm PS particles + U87 glioma cells and RBCs	90% ± 2.4% of U87 glioma cells could be isolated from the RBCs.
Shi, Jinjie et al. [75]	2011	SSAW	2 single-electrode IDTs Width = 25 μm Spacing = 25 μm Finger pairs = 20	100	Channel with 1 inlet and 1 outlets Dimensions: height = 100 μm; width = 50 μm Flow rate = 7 μL/min	3D focusing (sheathless)	Microscale: 1.9 μm PS particles	The particles are focused laterally and vertically.
Nam, Jeonghun et al. [51]	2012	SSAW	2 single-electrode IDTs Width = 250 μm Spacing = 250 μm	1000	Channel with 3 inlets and 5 outlets Dimensions: height = 200 μm; width = 300 μm Flow rate: sample = 8 μL/min; sheath flow = 16 μL/min	Separation by density	Microscale: 150.7 ± 11.3 μm cells encapsulated in alginate beads	More dense beads were collected with a recovery rate of over 97% and a purity of over 98% at a rate of 2300 beads per minute with acceptable cell viability.
Guldiken, Rasim et al. [76]	2012	SSAW	4 single-electrode IDTs Width = 75 μm Spacing = 75 μm Finger pairs = 20	300	Channel with 1 inlet and 3 outlets Dimensions: height = 100 μm; width = 150, 300 μm Flow rate = [0.5–5] μL/min	Focusing and sheathless separation	Microscale: 3, 5, and 10 μm PS particles	Separation of 100% for 10 μm and 94.8% for 3 μm particles (with the lowest flow rate).
Ai, Ye et al. [13]	2013	SSAW	2 single-electrode IDTs Width = 75 μm Spacing = 75 μm Finger pairs = 20 Length = 9 mm	300	Channel with 3 inlets and 3 outlets Dimensions: height = 25 μm; width = 120 μm; length = 15 mm Flow rate: sample = 0.2 (PS) and 0.5 μL/min (bacteria); sheath flow = 0.8 (PS) and 4 μL/min (bacteria)	Separation	Microscale: 1.2, 5.86 μm synthetic microspheres + *Escherichia coli* bacteria, peripheral blood mononuclear cells	The purity of separated *E. coli* bacteria was 95.65%
Jo, Myeong Chanand and Guldiken, Rasim [77]	2014	SSAW (phase shift)	4 single-electrode IDTs Width = 75 μm Spacing = 75 μm Finger pairs = 25	300	Channel with 1 inlet and 3 outlets Dimensions: height = 100 μm; width = 150 μm	Manipulation	Microscale: 5 μm PS particles	The particle displacement changed almost linearly as a function of the phase-shift.
Ding, Xiaoyun et al. [11]	2014	taSSAW	2 single-electrode IDTs Width = 50 μm Spacing = 50 μm Finger pairs = 24 Tilted angle = 15° Total aperture = 4 mm	200	Channel with 3 inlets and 2 outlets Dimensions: height = 75 μm; width = 1 μm; length = 4 mm Flow rate: sample (PS) = 1.50 mm/s; sample (MCF-7) = 2 μL/min	Separation	Microscale: 2, 7.3, 9.9 and 10 μm PS particles + MCF-7 breast cancer cells and WBCs	Separation efficiency of 97% for 9.9 μm and 7.3 μm and 99% for 2 and 10 μm. MCF-7 recovery rate of 71% and purity of 84%.
Lee, Kyungheon et al. [78]	2015	SSAW	2 single-electrode IDTs Width = 25 μm Spacing = 25 μm Length = 5.2 mm	100	Channel with 3 inlets and 3 outlets Dimensions: height = 80 μm; width = 60 μm Flow rate = 2.8 mm/s	Separation	Nanoscale: 190 nm, 1000 nm PS particles + Exosomes, larger microvesicles + microvesicles, red blood cells	> 90% separation yields (PS particles) + recovery rate > 80% for exosomes and > 90% for microvesicles.
Li, Peng et al. [12]	2015	taSSAW	2 single-electrode IDTs Width = 50 μm Spacing = 50 μm Length = 10 mm Tilted angle = 5°	200	Channel with 3 inlets and 2 outlets Dimensions: height = 110 μm; width = 800 μm; length = 10 mm Flow rate: sample = 20 μL/min; sheath flow = 50 μL/min	Separation	Microscale: CTCs (average diameters of 16 or 20 μm) and WBCs (∼12 μm)	Cancer cell recovery rate was > 83% (83–96%) and WBC removal rate was ∼90%.
Guo, Feng et al. [59]	2016	SSAW (phase shift)	4 single-electrode IDTs (Two pairs arranged orthogonally to each other) Width = 75 μm Spacing = 75 μm Finger pairs = 40 Total aperture = 1 cm	300	Chamber (static sample) Dimensions: height = 100 μm; width = 1.8 mm; length = 1.8 mm	3D trapping and manipulation	Microscale: 1, 4.2, 7.3, and 10.1 μm PS particles + 3T3 mouse fibroblast, HeLa S3 cellse	Manipulation of a single cell or particle placing it at a desired location with 1 µm accuracy in the x–y plane and 2 µm accuracy in the z direction.
Mao, Zhangming et al. [34]	2016	SSAW	2 single-electrode IDTs Width = 75 μm Spacing = 75 μm Finger pairs = 30	300	Channel with 1 inlet and 1 outlet Dimensions: height = 60 μm; width = 170, or 340 μm Flow rate = 10 μL/min	Study manipulation in narrow channels	Microscale: 10.11 μm PS and PDMS particles	The 2D SSAW microfluidic model developed matched the experiments, whereas the 1D harmonic standing waves model failed in the predictions.
Wu, Mengxi et al. [79]	2017	taSSAW	2 IDTs with floating electrodes Width = 10 μm Spacing = 10 μm Finger pairs = 80 Tilted angle = 15°	120	Channel with 2 inlets and 2 outlets Dimensions: height = 100 μm; width = 800 μm Flow rate: sample = 4 μL/min; sheath flow = 12 μL/min	Deflection and separation	Nanoscale: 110, 220, 240, 500, 600, 700 and 900 nm PS particles	For 900 and 600 nm particles, the removal rates were 96.6% and 80.4%, respectively. The recovery rates for 220 and 110 nm particles were 85.6% and 90.7%.
Lee, Junseok et al. [50]	2017	SSAW (phase shift)	2 single-electrode IDTs Width = λ/4 Spacing = λ/4 Finger pairs = 23	285	Channel with 2 inlets and 2 outlets Dimensions: height = 80 μm; width = 1050 μm Flow rate: sample = 5 μL/min; sheath flow = 5 μL/min	Separation	Microscale: 2, 6, and 12 μm PS particles + Human Keratinocytes (HaCaT)	Cell-bead mixture (2 μm PS and HaCaT) was separated with 83% efficiency.
Simon, Gergely et al. [35]	2017	SSAW (phase shift)	2 single-electrode IDTs Width = 75 μm Spacing = 75 μm Finger pairs = 20	300	Channel with 3 inlets and 2 outlets Dimensions: height = 50 μm; width = 240 μm; length = 2 cm	Separation	Microscale: 4.5, 5, 6, 10, and 15 μm PS particles	Efficiency of 90% (separating 10–15 µm) and 75% (separating 5–6 µm).
Li, Sixing et al. [15]	2017	taSSAW	2 single-electrode IDTs Width = 50 μm Spacing = 50 μm Tilted angle = 15°	200	Channel with 3 inlets and 2 outlets Dimensions: height = 75 μm; width = 1000 μm Flow rate: sample = 0.5 or 1 μL/min; sheath flow = 7 and 9 μL/min	Separation	Microscale: *Escherichia coli* bacteria (2 μm × 0.25–1.0 μm), RBCs (≈6.2–8.2 μm) and human blood samples	*E. coli* was separated from RBCs with a purity of more than 96%.
Nguyen, Tan Dai et al. [80]	2020	SSAW (phase shift)	4 single-electrode IDTs (two pairs arranged orthogonally to each other) Width = 75 μm Spacing = 75 μm Finger pairs = 60 Total aperture = 1 cm	300	Chamber (static sample) Dimensions: height = 100 (2D), 1000 (3D) μm; width = 1.5 mm; length = 1.5 mm	3D manipulation	Microscale: 10, 20 μm PS particles + Breast cancer cells (MCF-7)	Successfully relocated targets to specified coordinates.
Qian, Jingui et al. [81]	2020	SSAW—Lamb waves	4 chirped IDTs (two pairs arranged orthogonally to each other) Width = [50–75] μm Finger pairs = 26 Total aperture = 4.4 cm	≈[210–285]	Chamber (static sample) Dimensions: height = 50 μm; diameter = 1000 μm; sidewall width = 1 mm	2D manipulation in a single-use microfluidic chamber	Microscale: 5, 9 and 13 μm PS particles	Successfully shifted the position of microbeads on the disposable microchamber.
Zhao, Shuaiguo et al. [14]	2020	taSSAW	2 IDTs with floating electrodes Width = 10 μm Spacing = 10 μm Finger pairs = 80 Tilted angle = 15°	120	Channel with 3 inlets and 2 outlets Dimensions: height = 75 μm; width = 800 μm Flow rate: sample = 2 μL/min; sheath flow = 2 and 6 μL/min	Deflection and separation	Micro- and nanoscale: 110 nm, 400 nm, 1 μm, 2, 4.5, 6, and 10 PS particles, 660 nm SiO_2_ and 200 nm Ag particles + *E. coli* and human RBCs	Separation purity of up to 96% when separating *E. coli* from human RBCs.
Simon, Gergely et al. [43]	2021	SSAW	2 single-electrode IDTs Width = 75 μm Spacing = 75 μm	300	Channel with 3 inlets and 2 outlets Dimensions: height = 50 μm; width = 240 μm Flow rate: sample = 0.15 μL/min; sheath flow = 0.4 and 0.5 μL/min	Sorting	Microscale: 1, 6, 10, and 14.5 μm PS particles + RBCs, white blood cells	Purity and efficiency coefficients above 75 ± 6% and 85 ± 9% (for particles with 6, 10, and 14.5 μm) + 78 ± 8% efficiency and 74 ± 6% purity (for blood cells and 1 μm particle).
Ng, Jia Wei and Neild, Adrian [82]	2021	Cascaded SSAW	3 pairs of single-electrode IDTs Width = 25 μm Spacing = 25 μm Total aperture = 1 mm IDTs pairs distance = 12.5 λ	100	Channel with 3 inlets and 4 outlets Dimensions: height = 25 μm; width = 200 μm Flow rate: sample = 5 μL/min; sheath flow = 2 μL/min	Sorting and separation controlling multiple particle trajectories	Microscale: 5 μm PS particles	The particles were sorted into four distinct outlets using different actuation combinations.
Liu, Guojun et al. [83]	2022	taSSAW	2 single-electrode IDTs Width = 50 μm Spacing = 50 μm Finger pairs = 50 Tilted angle = 15°	200	Channel with 3 inlets and 3 outlets Dimensions: height = 100 μm; width = 500 μm Flow rate (for the authors’ best results): sample = 1.39 μL/min; sheath flow = 3.47 and 13.89 μL/min	Separation by density	Microscale: 10 μm PS particles, PMMA and SiO_2_ microspheres	Separation rates were 94.86%, 92.21%, and 90.36%, and the separation purities were 93.24%, 89.07%, and 91.81% for SiO_2_, PMMA, and PS particles, respectively.
Hsu, Jin-Chen and Chang, Chih-Yu [84]	2022	SSAW—Lamb waves	2 single-electrode IDTs Width = 100 μm Spacing = 100 μm Finger pairs = 20 Total aperture = 4.4 mm	400	For aggregation: channel with 2 inlets and 2 outlets; dimensions: height = 60 μm; width = 500 μm Flow rate = 4 μL/min For separation: channel with 3 inlets and 2 outlets; dimensions: height = 60 μm; width = 300 or 600 μm Flow rate: sample = 3 μL/min; sheath flow = 1 μL/min	Aggregation and separation	Microscale: 2, 7, and 10 μm PS particles	Separation of 2 and 10 μm particles with a manual estimation of the separation rate > 90%.
Sachs, Sebastian et al. [85]	2022	SSAW	2 single-electrode IDTs (3 different configurations) Width Spacing = 20 or 90 or 150 μm Finger pairs = variable Total aperture = 2 mm	20 or 90 or 150	Channel with 1 inlet and 1 outlet Dimensions: height = 85 or 185 or 530 μm; width = 500 μm; length = 800 μm Flow rate = [0.15–4] μL/min	To study channel height and SAW wavelength impact on separation	Micro- and nanoscale: 0.55, 1.14, μm PS particles	The numerical model optimized based on experimental data showed that shallow microchannels and large wavelengths are advantageous for particle separation.
Han, Junlong et al. [42]	2023	taSSAW	2 single-electrode IDTs Width = 44 μm Spacing = 44 μm Tilted angle = 15° Total aperture = 6 mm	175	Channel with 3 inlets and 2 outlets Dimensions: height = 100 μm; width = 800 μm; length = 8 mm Flow rate: sample = 2 μL/min; sheath flow = 2 and 6 μL/min	Separation	Micro- and nanoscale: 1, 3, 6 μm, and 100 nm PS particles + 750 nm SiO_2_ particles	In the different experiments, the separation purity was 80 to 95% and the separation efficiency (yield) was 83 to 97%. (depending on size and flow rate).
Liu, Xia et al. [86]	2023	SSAW	2 single-electrode IDTs Width = 75 μm Spacing = 75 μm Finger pairs = 20	300	Channel with 1 inlet and 1 outlet (static sample) Dimensions: height = 70 μm; width = 500 μm	Study precise 3D motion control	Micro- and nanoscale: 0.5, 5, and 10 μm PS particles	They demonstrated that micro- and nanoparticles can move in three dimensions when acoustic radiation force and acoustic streaming interact.

Table abbreviations: taTSAW, tilted-angle traveling surface acoustic waves; taSSAW, tilted-angle standing surface acoustic waves; 3D, three-dimensional; PS, polystyrene; RBCs, red blood cells; SiO_2_, silicon dioxide; Ag, silver; *E. coli*, *Escherichia coli*; NA, not applicable; APM, acoustic plate mode.

## Data Availability

All generated/compiled data are contained within the article itself.

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
