# Peer review of "A Review of SAW-Based Micro- and Nanoparticle Manipulation in Microfluidics"

_sensors, 2025, doi:10.3390/s25051577_

Round 1

Reviewer 1 Report

Comments and Suggestions for Authors

Thank you for a very interesting, well-organized and comprehensive review. I feel that provided table, which is the core part of the paper, might be really useful for both the ongoing researchers and those who are new to the topic. Having no doubts about potential paper publication, I want to mention just a couple of possible misprints:

1) line 115. Probably it should be "thantalate"
2) Fig. 1, a). Usually the total aperture is considered as the length where two consecutive fingers overlay (not literally of course), but in this figure it shows the whole length of the finger. Please check, if it is really what you wanted to specify.

  Here are the detailed comments:   • What is the main question addressed by the research? The paper represents a review of different techniques utilizing SAWs for particle manipulation in microfluidics. It does not contain novel results but summarizes and analyzes previously published results and concepts.

• Do you consider the topic original or relevant to the field? Does it
address a specific gap in the field? Please also explain why this is/ is not
the case.
Once again, the paper is a review and it is not obliged to represent some novel materials, but should give a detailed explanation of the current status of the topic along with deep analysis of trends and future prospects. And the paper does this with no doubts.
• What does it add to the subject area compared with other published
material?
It provides a detailed explanation of the current status of the topic along with deep analysis of trends and future prospects. 
• What specific improvements should the authors consider regarding the
methodology?
The methodology is fine
• Are the conclusions consistent with the evidence and arguments presented
and do they address the main question posed? Please also explain why this
is/is not the case.
Conclusions of the review paper should contain information about existing or arising trends in the area and/or underline the current level of the technology. The paper under consideration, for example, gives the percentage of papers considering different SAW modes, specifies the average level of device efficiency (in % of the particles filtered), the use of LiNbo3 in all the reviewed sources, and so on. Thus, I assume that the conclusion addresses the main question posed. 
• Are the references appropriate? The references are appropriate.
• Any additional comments on the tables and figures. I have no additional comments on the tables and figures.

Author Response

Reviewer 1

Thank you for a very interesting, well-organized and comprehensive review. I feel that provided table, which is the core part of the paper, might be really useful for both the ongoing researchers and those who are new to the topic. Having no doubts about potential paper publication, I want to mention just a couple of possible misprints:

  1. line 115. Probably it should be "thantalate"

Answer: Thank you for pointing this out. We have made the necessary correction.

  1. 1, a). Usually the total aperture is considered as the length where two consecutive fingers overlay (not literally of course), but in this figure it shows the whole length of the finger. Please check, if it is really what you wanted to specify.

Answer: Thank you for your observation. We have reviewed and corrected the figure to accurately represent the total aperture as intended.

Reviewer 2 Report

Comments and Suggestions for Authors

The manuscript "A review on SAW-based micro and nanoparticle manipulation in microfluidics" focuses on a vibrant and urgent field of microfluidics that is combinations of microfluidic devices with acoustic waves. This field is of high research interest with definite potential applications in a variety of lab-on-chip fields such as particle manipulation, separation, or analysis.

The review is thoroughly performed and the authors provided relevant and up-to-date references that describe recent research achievements in this new multidisciplinary field.

The work can be further considered for publication after the authors consider the following comments and concerns:

  1. To fit the scope of Sensors, the manuscript will benefit from discussing sensing capabilities of SAW microfluidics for example, with respect to nanoparticles, microparticles or chemical agents. In its current state, the manuscript relates indirectly to SAW sensors only in page 3.
  2. In Section 3.1, the governing equations are standalone. Is there any possible combinations of equations that characterize fluid dynamics and acoustic wave propagation or piezoelectricity? Such equations will be of great interest to scientists who perform research in SAW microfluidics.
  3. The core of the review is Table 1, which occupies 8 pages. Although the authors did a great job by summarizing SAW-based devices, this table shows microfluidic devices for manipulation. Particle manipulation relates indirectly to sensing, so the revised manuscript will benefit from discussing such a relation.
  4. It is not clear why Fig. 4 was shown in Discussion. This figure represents various configurations of SAW microchips. It is informative and summarizing and fits better Section 2. Please clarify.
  5. In Line 45, the authors mentioned a growth in the development of SAW microfluidic chips. The review, therefore, will benefit from adding an image of a real SAW microfluidic device, if possible.

Minor comment:

1. The text in Lines 410-411 is not quite clear. Does it represent abbreviations?

Author Response

Reviewer 2

The manuscript "A review on SAW-based micro and nanoparticle manipulation in microfluidics" focuses on a vibrant and urgent field of microfluidics that is combinations of microfluidic devices with acoustic waves. This field is of high research interest with definite potential applications in a variety of lab-on-chip fields such as particle manipulation, separation, or analysis.

The review is thoroughly performed and the authors provided relevant and up-to-date references that describe recent research achievements in this new multidisciplinary field.

The work can be further considered for publication after the authors consider the following comments and concerns:

  1. To fit the scope of Sensors, the manuscript will benefit from discussing sensing capabilities of SAW microfluidics for example, with respect to nanoparticles, microparticles or chemical agents. In its current state, the manuscript relates indirectly to SAW sensors only in page 3.

Answer: Thank you for your suggestion. We did not discuss the sensors’ applications in depth as it would make the manuscript too long, and our focus was the particles’ manipulation. However, we have addressed this by adding a short paragraph in the introduction (lines 58-65) discussing the sensing capabilities of SAW in microfluidic domains. We believe this addition, although summarized, helps to strengthen the manuscript’s alignment with the scope of Sensors.

  1. In Section 3.1, the governing equations are standalone. Is there any possible combinations of equations that characterize fluid dynamics and acoustic wave propagation or piezoelectricity? Such equations will be of great interest to scientists who perform research in SAW microfluidics.

Answer: We appreciate the reviewer’s suggestion regarding the presentation of the governing equations in Section 3.1. We have carefully considered different ways to structure this section but found that the current approach was probably the one that provides the best clarity while maintaining accessibility for a broad audience. In particular, most commercial softwares for simulation (for instance COMSOL Multiphysics, where the authors usually work), also solve this problem using the fluid dynamics, acoustic wave propagation and piezoelectricity equations by themselves. What happens is that, in a Multiphysics model like these, the pressure outputted by the piezoelectric domain enters as a pressure input in the acoustic propagation and in the fluid dynamics equations. But no new equations enter the model. Therefore, we have decided to keep the equations, but added small details in the text to explain how these variables relate in the different equations.

  1. The core of the review is Table 1, which occupies 8 pages. Although the authors did a great job by summarizing SAW-based devices, this table shows microfluidic devices for manipulation. Particle manipulation relates indirectly to sensing, so the revised manuscript will benefit from discussing such a relation.

Answer: Thank you for the observation regarding the relevance of particle manipulation to sensing. To address this, we added some paragraphs in the text in order to complement the discussion about SAW sensors with applications in microfluidics. In this context, we highlight the inclusion of a paragraph in the manuscript explaining how SAW-based particle manipulation techniques contribute to sensing applications (lines 485-490).

  1. It is not clear why Fig. 4 was shown in Discussion. This figure represents various configurations of SAW microchips. It is informative and summarizing and fits better Section 2. Please clarify.

Answer: We appreciate your suggestion. However, we believe that former Figure 4 (current figure 7) provides a visual summary that helps contextualize and summarize, with some examples, the data presented in table 2 and supports the analysis in Section 5. Therefore, we have decided to keep it in the Discussion section to maintain consistency and facilitate a better understanding of the key points discussed. Nevertheless, to comply with the Reviewer comment, we decided to include an image of a real SAW microfluidic device in Section 2 to further enhance the manuscript, and added a few more figures to help understand better the phenomena and devices discussed.

  1. In Line 45, the authors mentioned a growth in the development of SAW microfluidic chips. The review, therefore, will benefit from adding an image of a real SAW microfluidic device, if possible.

Answer: Thank you for pointing this out. We agree that including an image of a real SAW microfluidic device enhances the review. Accordingly, we have added an image to better illustrate the development and practical implementation of these devices (section 2, Figure 1).

Minor comment:

  1. The text in Lines 410-411 is not quite clear. Does it represent abbreviations?

Answer: Thank you for your feedback. Yes, the text represents abbreviations used in the table. For clarity, we have added a note at the beginning stating, "Table abbreviations:" to ensure readers can easily understand.

Reviewer 3 Report

Comments and Suggestions for Authors

sensors-3434177-peer-review-v1
A review on SAW-based micro and nanoparticle manipulation in microfluidics

Surface acoustic waves (SAWs) are conventionally utilized for design passive RF filters and resonators. Then, for a long time, SAWs have been developed for potential applications in microfluidics for manipulating micro-/nano-particles in biomedical related applications. Herein, this manuscript reviews the SAW-based micro and nanoparticle manipulation in microfluidics. Below are a few comments.

1.    In the Introduction section, I fail to see the development of SAW based micro/nano-particle manipulation. The authors are suggested to include more detailed discussion on the current main progress and challenges.
2.    Though the authors use a comparison Table to include main results about the micro/nano-particle manipulation, it is still required to include more Figures to discuss different conditions, e.g., different kind of particles, particles with different sizes, particles with different functions, different specific applications.
3.    Following the 2nd comment, the number of references is around 100, however, there are just four figures in the whole manuscript. This is not enough at all.
4.    Besides, the authors discuss a few parameters that affecting the manipulation of particles. It is then suggested to discuss the influences of different parameters in separate Sub-Sections.

Author Response

Reviewer 3

Surface acoustic waves (SAWs) are conventionally utilized for design passive RF filters and resonators. Then, for a long time, SAWs have been developed for potential applications in microfluidics for manipulating micro-/nano-particles in biomedical related applications. Herein, this manuscript reviews the SAW-based micro and nanoparticle manipulation in microfluidics. Below are a few comments.

  1. In the Introduction section, I fail to see the development of SAW based micro/nano-particle manipulation. The authors are suggested to include more detailed discussion on the current main progress and challenges.

Answer: We appreciate the reviewer’s suggestion. In response, we have revised the Introduction to include a more detailed discussion of recent advancements and challenges in the field (lines 90-101).

  1.    Though the authors use a comparison Table to include main results about the micro/nano-particle manipulation, it is still required to include more Figures to discuss different conditions, e.g., different kind of particles, particles with different sizes, particles with different functions, different specific applications.
    3.    Following the 2nd comment, the number of references is around 100, however, there are just four figures in the whole manuscript. This is not enough at all.

Answer: Thank you for your valuable feedback. Following your suggestions, we have added more figures to enhance the manuscript. In Section 2, we included an image of a real SAW microfluidic device. In Section 3, we added a schematic diagram illustrating the forces acting on the particles, as well as an image of two real devices with the respective particle patterning. We believe these additions will help provide a clearer understanding of the different conditions and applications discussed in the manuscript.

  1.    Besides, the authors discuss a few parameters that affecting the manipulation of particles. It is then suggested to discuss the influences of different parameters in separate Sub-Sections.

Answer: We appreciate the reviewer’s suggestion to discuss the influence of different parameters in separate sub-sections. As suggested, we structured our manuscript accordingly, where each key parameter affecting particle manipulation is discussed in its respective sub-section.

Reviewer 4 Report

Comments and Suggestions for Authors

This manscript make a review on SAW-based micro and nonoparticle manipulation in microfluidics. It has much significance. But for a review, one of the most important thing is the summary classifiction for the existing studies. This manuscript is more like a draft. I have some advices to revise:

  1. In Page3, label in the figure 1 is wrong, "the total aperature"is the overlap of the positive and negative electrodes.
  2. In Page 5, after introduing IDT structures, authors can make a summary for the application for different IDT structure.
  3. In Page 6, formats of the euations (2) and (3) are not irregularity. Also forms of the equations of (8)/(9)/(10) are not irregularity. 
  4. In part 4, authors can make a schematic diagram for the acoustic streaming and acoustic radation force on the particles.
  5. In the table, authors should make a major change: 1) the information of the authors can change to "team"; 2) it is better to offer digram for the information of the IDT; 3)the table is just listed now, it shold be sumarized according to the application or other subject; 4) furthermore, the list can be combined according to the similiar design from the same team; 5) the working frequency can be replenished.
  6. Is the figure 4 from the other reference? If this, do the authors process the problem of the copyright?
  7. In the part 5, it may be better to make a classification summary to discussion using subheading.

Comments on the Quality of English Language

The overall of the quality of the Enlish is good, but it can be improved in many details. Such as, in the line 422 in Page 17, it is not standard to give short form as it is not the first time to appear. It will be better to polish the whole manscript.

Author Response

Reviewer 4

This manuscript makes a review on SAW-based micro and nanoparticle manipulation in microfluidics. It has much significance. But for a review, one of the most important things is the summary classification for the existing studies. This manuscript is more like a draft. I have some advices to revise:

  1. In Page3, label in the figure 1 is wrong, "the total aperture" is the overlap of the positive and negative electrodes.

Answer: Thank you for pointing this out. We have made the necessary correction.

  1. In Page 5, after introducing IDT structures, authors can make a summary for the application for different IDT structure.

Answer: Thank you for your suggestion. We have added a summary table after introducing the different IDT structures. This table provides a concise overview of the key features and applications of each IDT design, making it easier for the reader to compare the different configurations and their respective uses in SAW-based systems.

  1. In Page 6, formats of the equations (2) and (3) are not irregularity. Also forms of the equations of (8)/(9)/(10) are not irregularity. 

Answer: We appreciate your feedback regarding the formatting of Equations (2), (3), (8), (9), and (10). We have carefully reviewed these equations for inconsistencies in notation, spacing, and alignment but could not identify any issues. If the problem persists, we would appreciate further clarification to ensure we can make the necessary adjustments.

  1. In part 4, authors can make a schematic diagram for the acoustic streaming and acoustic radiation force on the particles.

Answer: We agree that a schematic diagram would be helpful. Following your recommendation, we have included such an image in Section 3.1, after the governing equations, to help the reader connect the physics concepts with the forces acting on the particles.

  1. In the table, authors should make a major change:

1) the information of the authors can change to "team";

Answer: Following your recommendation, we have made the requested change and now refer to the authors’ information as "team" in the table.

2) it is better to offer diagram for the information of the IDT;

Answer: We appreciate your suggestion and agree that a diagram could improve clarity. However, to maintain a more concise presentation, and to not extend the length of the paper, we have decided to keep the current format. For each case, we describe the type of IDT, which the readers can consult in Figure 3 to understand their format. Additionally, we have ensured consistency by using the same terminology in the table as in Figure 3, where the different IDT designs are presented.

3)the table is just listed now, it should be summarized according to the application or other subject; 4) furthermore, the list can be combined according to the similar design from the same team;

Answer: Thank you for your suggestion. The table, right now, is structured based on SAW type, with TSAW studies listed first, followed by SSAW studies, both organized chronologically by year. This arrangement provides a clear timeline of developments within each SAW category. We understand we could summarize the data according to application, similar design or another relevant criterion, and consider the type of SAW waves a proper one.

 5) the working frequency can be replenished.

Answer: We appreciate your comment. However, instead of including the working frequency, we have chosen to present the wavelength, as this is the parameter most commonly reported in the referenced studies (frequency is absent in the descriptions in many of the works). While the SAW frequency could be theoretically calculated, it may correspond to non-accurate values of the operating frequency of the device (depending on the material, configuration, etc). Therefore, as wavelength is reported in all studies, and assures an accurate description of the wave propagation, we have decided to maintain the wavelength information instead of the frequency.

  1. Is the figure 4 from the other reference? If this, do the authors process the problem of the copyright?

Answer: Thank you for your question. Figure 4 is a compilation of several figures from other articles for which we have obtained permission to use. However, the compilation itself is original and authored by us. We have ensured that the necessary permissions were obtained for the individual figures as included in the figure caption (current Figure 7).

  1. In the part 5, it may be better to make a classification summary to discussion using subheading.

Answer: Thank you for your suggestion. To enhance readability and improve comprehension, we have structured section 5 into sub-sections.

Comments on the Quality of English Language

The overall of the quality of the English is good, but it can be improved in many details. Such as, in the line 422 in Page 17, it is not standard to give short form as it is not the first time to appear. It will be better to polish the whole manuscript.

Answer: Thank you for your feedback. We have corrected the issue in former line 422 (currently line 503) and have thoroughly revised the entire manuscript.

Round 2

Reviewer 2 Report

Comments and Suggestions for Authors

In the revised manuscript, the authors addressed my comments and concerns. 

The major comment about a correlation between possible sensing applications and SAW-based microchips was adressed in the revision. Also, the revision benefited from a demonstration of a real SAW-based microchip (Fig. 1).

The revised review can be further considered for publication.

Reviewer 3 Report

Comments and Suggestions for Authors

The authors have properly addressed the comments and therefore the manuscript should be acceptable for publication in its current form now.

Reviewer 4 Report

Comments and Suggestions for Authors

Most of the comments have been modified by the authors. But in the Table 2, not just the title change to "team". It should be carefully sorted for the referenced according to the same team. Also, in the line 61 Page 2, the citation has made a wrong format.

Comments on the Quality of English Language

It may be polished by the professional staff.